# Discontinuities in Wintertime Warming in Northern Europe during 1951–2016

**Mikhail M. Latonin [1,2,*], Vladimir A. Lobanov [3] and Igor L. Bashmachnikov [1,2]**

[1]   Climate Group, Nansen International Environmental and Remote Sensing Centre, 14th Line 7, Office 49, Vasilievsky Island, 199034 Saint Petersburg, Russia; igorb1969@mail.ru

[2]   Department of Oceanography, Saint Petersburg State University, Universitetskaya Emb. 7-9, 199034 Saint Petersburg, Russia

[3]   Department of Meteorology, Climatology and Environmental Protection, Russian State Hydrometeorological University, Voronezhskaya street 79, 192007 Saint Petersburg, Russia; lobanov@EL6309.spb.edu

\*   Correspondence: mikhail.latonin@niersc.spb.ru; Tel.: +7-905-283-07-96

**Abstract:** Although there is a general consensus about the trends of current climate change, the North Atlantic region deserves special attention, as it is the key region for many climate processes. The aim of this study is to assess the climatic changes in this region for the period 1951–2016, based on the analysis of surface air temperature (SAT) observations from weather stations, and the North Atlantic Oscillation (NAO). Statistical modeling of time series for January, February and March shows a stepwise increase of SAT in Northern Europe in 1987–1989, with the stationarity increasing towards spring. The divided trends of the NAO and SAT indicate a good coherence at the level of climate tendencies. This research reveals the discontinuity of the present-day wintertime warming in Northern Europe, with a warming of about 2 °C after the stepwise jump of the SAT.

**Keywords:** climate change; Northern Europe; North Atlantic; Greenland; triggered mechanism; trends; discontinuity of warming; SAT; NAO; winter climate

---

## 1. Introduction

The decades-long increase of the global surface temperature, despite a hiatus in global warming [1], is a well-documented feature of present climate change, as reflected, for example, in the global land–ocean temperature index [2] and a number of publications, such as [3,4]. The previous warming period of the 1920s–1940s, called the early twentieth-century warming (ETCW), was mainly confined to the high northern latitudes, and was thus different from the current situation [5]. Although the mechanisms involved during that time are not fully understood, there is a general agreement in the scientific community that it was caused by long-term natural variability [6–8]. Currently, both natural and anthropogenic influences are considered as the drivers of the present-day warming [9,10]. The patterns of the current climate change are shown in an IPCC Special Report on the impacts of global warming of 1.5 °C above pre-industrial levels [11]. For instance, Figure 1.3 of the report depicts that the Arctic region is most influenced by the warming. This phenomenon is called Arctic amplification, a regional manifestation of a more general phenomenon; namely, polar amplification [12]. This is a highly debated research area; however, a general physical understanding of this phenomenon lies in a higher sensitivity of polar regions to external radiative forcings [13], an interplay of climate feedbacks at different latitudes [14,15], and variations in meridional atmospheric and oceanic heat transport [16–18]. These peculiarities are concentrated near the surface; therefore, surface air temperature (SAT) records are one of the clues that should be used to explain the pronounced surface

climate warming. This environment is the habitat in which the most infrastructure is located. Therefore, a thorough understanding of the processes involved is required to ensure sustainable development.

The North Atlantic region is essential for understanding many present climate processes, because it is the location of an atmospheric center of action, the Icelandic Low [19]. Furthermore, the Greenland Ice Sheet has started to experience substantial ice melting, which may cause an acceleration of sea level rise [20–25]. The North Atlantic Oscillation (NAO), a regional manifestation of the Arctic Oscillation (AO) [26], is the leading mode of atmospheric variability affecting weather and climate in the North Atlantic region, including North America and Europe [27]. For instance, an extremely negative phase of NAO during the winter of 2009–2010 was connected with the very cold winter in Europe, as well as with heavy snowfalls, and record-breaking low temperatures along the eastern coast of North America [28], which threatened life and led to unusually high energy demand [29]. It is worth noting that the NAO effect on weather and climate is not just event-specific; long-term climate tendencies are also affected by this mode of variability, which is documented in many studies, e.g., [30–34]. Another essential mode of variability in the North Atlantic region is the Atlantic Multidecadal Oscillation (AMO), which characterizes large-scale multidecadal variations of the sea surface temperature (SST) over the North Atlantic basin, usually over 0–80° N [35,36]. This climate index has various impacts. For example, it has been shown in a number of studies that the AMO modulates mean SAT in the Northern Hemisphere [32,37] and affects the Arctic sea ice [38–41].

Although the NAO index is driven mainly by internal variability in the atmosphere, which is also confirmed by unforced model simulations [33], the external forcings also play an important role. There is evidence that they might contribute to the shift towards the positive polarity of the NAO [42]. This is indirectly supported by the average winter NAO pattern during the modern era (1965–1995), which shows the highest positive trend for the instrumental record [33]. This period nearly coincides with the unprecedented reduction in the Atlantic meridional overturning circulation (AMOC) and accelerated surface temperature warming in 1975–1995 [43,44]. This combination is a paradox, indicating an unusual behavior of the climate system and an increased likelihood of hidden tipping points; therefore, it is highly relevant to continue an in-depth investigation of the time series during the present period.

Overall, the SAT, climate indices and circulation patterns are essential indicators that can be used to monitor and follow the present climate change. Despite numerous studies of their role in global and regional climates, more reliable information about their particular effects on climate tendencies is needed. In particular, although the periods of warming and cooling are known, whether their interaction produces a continuous warming signal or results in substantial interruptions for certain regions due to abrupt changes in the circulation patterns has not been widely investigated. This paper aims at explaining the climatic changes in terms of the different states of climate captured by the time series models applied to the SAT, and the subsequent relationships of the obtained results with the NAO. The possible interconnection with the AMO is not considered in this study because the length of the time series is not sufficiently long for the AMO period of 60–80 years, including the AMO index itself [45]. A detailed analysis of the AMOC indices is also beyond the scope of this study.

## 2. Materials and Methods

For the analysis, monthly SAT data for January, February and March were selected at 17 meteorological stations, the locations of which are shown in Figure 1.

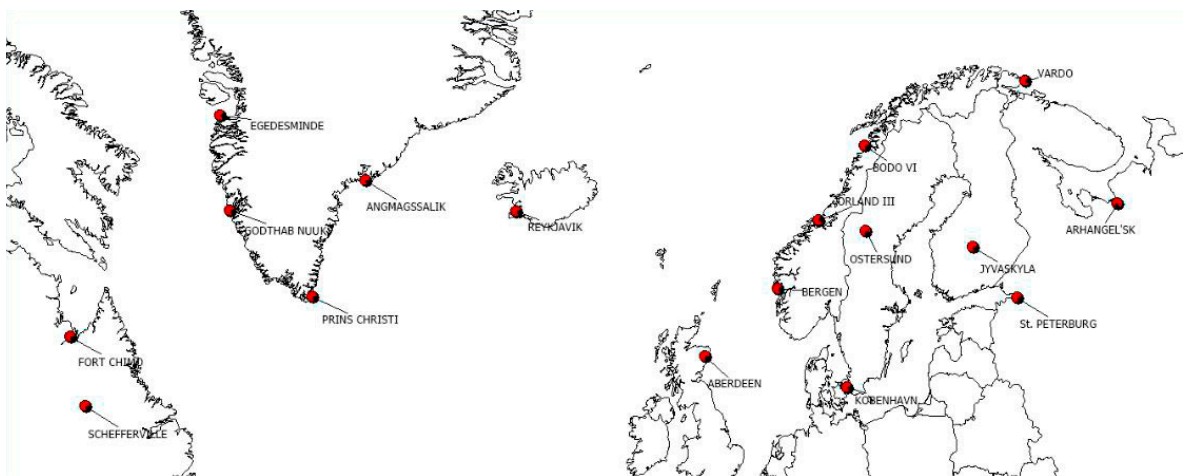

**Figure 1.** Location of weather stations.

These stations were selected taking into account the coverage of the area studied and the number of missing values in the datasets. Information was obtained from the international center of climate data at the Royal Netherlands Meteorological Institute (KNMI) [46]. The data were selected only for three months, because the focus of this research is on the patterns of the typical Arctic winter season (January–March), based on the fact that the maximum Arctic sea ice extent is achieved in March [47].

Geographically, all weather stations can be divided into four groups:

1. Eastern Canada;
2. Southern Greenland;
3. Northern Europe, with mainly marine influence;
4. Northern Europe, with both marine and continental influence, including the northwest of Russia.

An analysis of the regional database has shown that the time series have different lengths; however, from the second half of the 20th century to the present, data are already nearly fully available at all of the meteorological stations. This is an important point for assessing current climate change. However, some meteorological stations have significantly longer records, and it will be shown how they are used to contrast the present climate change with what was previously observed. A summary of the considered meteorological stations is given in Table 1.

**Table 1.** The length of the surface air temperature (SAT) time series at the meteorological stations in the North Atlantic region.

| Subregion | Name of the Meteorological Station | Range of Years | Number of Years | Latitude, ° N | Longitude, ° E | WMO Code |
|---|---|---|---|---|---|---|
| Eastern Canada | SCHEFFERVILLE | 1948–2016 | 69 | 54.80 | −66.80 | 71828 |
|  | FORT CHIMO | 1941–2016 | 76 | 58.10 | −68.40 | 71906 |
|  | EGEDESMINDE | 1949–2016 | 68 | 68.70 | −52.75 | 4220 |
| Southern Greenland | GODTHAB NUUK | 1866–2016 | 151 | 64.17 | −51.75 | 4250 |
|  | PRINS CHRISTI | 1949–2016 | 68 | 60.05 | −43.17 | 4390 |
|  | ANGMAGSSALIK | 1895–2016 | 122 | 65.60 | −37.63 | 4360 |
|  | REYKJAVIK | 1901–2016 | 116 | 64.13 | −21.90 | 4030 |
| Northern Europe with mainly marine influence | VARDO | 1829–2016 | 188 | 70.37 | 31.10 | 1098 |
|  | BODO VI | 1868–2016 | 149 | 67.27 | 14.37 | 1152 |
|  | ORLAND III | 1951–2016 | 66 | 63.70 | 9.60 | 1241 |
|  | BERGEN | 1816–2016 | 201 | 60.40 | 5.30 | 1317 |
|  | ABERDEEN | 1871–2016 | 146 | 57.20 | −2.22 | 3091 |
| Northern Europe with both marine and continental influence | KOBENHAVN | 1768–2016 | 249 | 55.68 | 12.55 | 6186 |
|  | OSTERSUND | 1949–2016 | 68 | 63.18 | 14.50 | 2226 |
|  | JYVASKYLA | 1950–2016 | 67 | 62.40 | 25.68 | 2935 |
|  | St. PETERBURG | 1743–2016 | 274 | 59.97 | 30.30 | 26063 |
|  | ARHANGEL'SK | 1813–2016 | 204 | 64.50 | 40.73 | 22550 |

As the main purpose of this research is to study present climate change, the SAT time series of all weather stations were considered primarily for the period 1951–2016. In this time range, there are very few data gaps, thus increasing the accuracy of the assessments. The remaining single gaps were recovered using NCEP/NCAR reanalysis data [48].

In addition, the data of a key climate index were used in this study; namely, the well-known NAO index, the station-based version of which was obtained from the NCAR climate data guide [49].

The core methodology applied here is the statistical modeling of the time series and an analysis of the trends divided based on that methodology. However, basic techniques are also applied: correlation matrices are used for the analysis of the spatial coherence and the significance of the differences in means is estimated by the two-sample Student's t-test [50,51]. The methodology described below for the time series modeling is well-known, and can be found in [50,52–56].

At any considered time interval, the structure of the time series can be represented by two main types of models: stationary and non-stationary. In a stationary model, the main parameters of the time series, such as the mean and variance, are constant over time, or stationary.

However, in addition to random fluctuations in the time series of climatic characteristics, non-stationary components may also occur due to the influence of climate system factors with large time scales. Two non-stationary models are considered in this study: stepwise climate change and the model of a linear trend.

The mechanism of stepwise changes, or the triggered mechanism, characterizes a nonequilibrium system which, for some time, can neutralize directed external influences or resist them until their combined effect transfers the system to a new level.

The model of stepwise changes is similar to two (or several) stationary models for two (or several) parts of the time series, which are characterized by a constant mean ($\overline{Y}$) and standard deviation ($\sigma$) for each part of the time series:

$$\overline{Y_1} = const1, \overline{Y_2} = const2, \sigma_1 = const1, \sigma_2 = const2. \tag{1}$$

The moment of stepwise changes is determined by iterations upon reaching the minimum value of the sum of the squared deviations of the two parts of the time series (S):

$$S = \sqrt{\frac{\sum\limits_{i=1}^{n_1} \left(Y_i - \overline{Y_1}\right)^2 + \sum\limits_{i=n_1+1}^{n_2} \left(Y_i - \overline{Y_2}\right)^2}{n - 1}} = min, \tag{2}$$

where $n_1$ and $n_2$ are the sizes of the two parts of the time series which constitute the original time series of size n. The minimum size of the first part of the time series is set to be $n_1 = 10$, for which $n_2 = n - n_1$, and then the size is increased: $n_1 = 11, 12, \ldots, k$, where $k = n - 10$ at which $n_2 = 10$. Once the minimum sum is found, the corresponding index is attributed to the year in the original time series ($T_{step}$). This procedure is related to the expanding window method.

The standard deviation of the residuals of the stepwise model for one step and two stationary intervals is given by:

$$\sigma_{step} = \sqrt{\frac{\sigma_1^2(n_1 - 1) + \sigma_2^2(n_2 - 1)}{(n_1 + n_2 - 1)}}, \tag{3}$$

where $\sigma_{step}$ is the standard deviation of the residuals of the stepwise model; $\sigma_1$ and $\sigma_2$ are the standard deviations of the stationary intervals of the time series; and $n_1$ and $n_2$ are the sizes of the stationary intervals.

The linear trend model is expressed by the following equation:

$$Y(t) = b_1 t + b_0, \tag{4}$$

where t is time, and $b_1$ and $b_0$ are the coefficients of the regression equation determined by the method of least squares:

$$b_1 = \frac{\sum\limits_{i=1}^{n} \left(Y_i - \overline{Y}\right)\left(t_i - \overline{t}\right)}{\sum\limits_{i=1}^{n} \left(t_i - \overline{t}\right)^2} \tag{5}$$

$$b_0 = \overline{Y} - b_1\overline{t} \tag{6}$$

The statistical significance of the linear trend is assessed based on the evaluation of the statistical significance of the correlation coefficient, R, given by:

$$R = \frac{\sum\limits_{i=1}^{n} \left(Y_i - \overline{Y}\right)\left(t_i - \overline{t}\right)}{\sqrt{\sum\limits_{i=1}^{n} \left(Y_i - \overline{Y}\right)^2 \sum\limits_{i=1}^{n} \left(t_i - \overline{t}\right)^2}} \tag{7}$$

The statistical significance of R is determined according to the condition R ≥ R*, where R* is a critical value of the correlation coefficient determined at a given number of the degrees of freedom (ν) and significance level (α), where ν = n − 2, n is the size of the time series and α = 5%.

For the linear trend model, the standard deviation of the residuals is calculated as follows:

$$\sigma_\varepsilon = \sigma_y \sqrt{1 - R^2}, \tag{8}$$

where $\sigma_y$ is the standard deviation of the given time series (the model of the stationary mean), $\sigma_\varepsilon$ is the standard deviation of the residuals relative to the linear trend model and R is a correlation coefficient of the linear trend equation.

To quantify the differences in the trend model and the model of stepwise changes from the model of the stationary mean, relative errors are calculated as follows:

$$\Delta_{tr.} = \left(\frac{\sigma_y - \sigma_\varepsilon}{\sigma_y}\right) * 100\% \tag{9}$$

$$\Delta_{step} = \left(\frac{\sigma_y - \sigma_{step}}{\sigma_y}\right) * 100\%, \tag{10}$$

where $\Delta_{tr.}$ and $\Delta_{step}$ are relative errors or differences (in percent) of the trend model and the model of stepwise changes from the model of the stationary sample. In this study, a difference of 10% and higher is considered to be significant.

To evaluate the statistical significance of monotonous (trend) and stepwise changes in the time series, the Fisher criterion is applied to show quantitatively if the differences between the residual variances of the chosen models and the variance of the stationary model are statistically significant. Fisher statistics for each of the two competing models with respect to the stationary model are calculated according to the following formulas:

$$F_{tr.} = \frac{\sigma_y^2}{\sigma_\varepsilon^2} \tag{11}$$

$$F_{step} = \frac{\sigma_y^2}{\sigma_{step}^2}. \tag{12}$$

Ideally, if the calculated values of the Fisher statistics are higher than their critical values, then the variances of the two models have a statistically significant difference, and the corresponding model (trend or stepwise changes) is statistically more effective than the model of the stationary sample.

In this study, the condition of 10%, applied to Equations (9) and (10), is considered sufficient for the trend model and the model of stepwise changes, respectively, to perform better than the model of the stationary sample.

## 3. Results

### 3.1. Correlations for the Period 1951–2016 Among All Weather Stations in January, February and March

To demonstrate the interconnection in the SAT among weather stations, thus identifying the regions with coherent changes, the correlation matrices were calculated for the considered months. For a better visual perception, the matrices were transformed to surfaces with cells, and color gradients with respective Pearson correlation coefficients were applied. The results for January are shown in Figure 2.

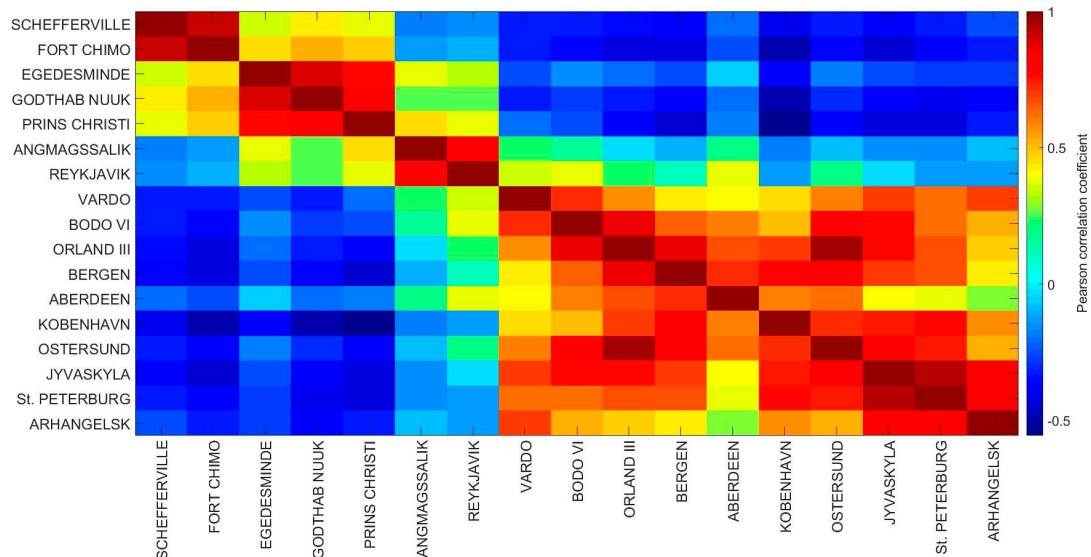

**Figure 2.** Pearson correlation coefficients among all weather stations in January (1951–2016 period).

As shown, the weather stations of Northern Europe have the highest correlations for multiple locations, and the SAT changes are unidirectional in this region. Moreover, all correlations among the stations from Vardo to Arhangelsk are statistically significant, and even between Aberdeen and Arhangelsk, with a value of 0.27. The only exception is the weather station in Reykjavik, which is expected as this weather station is isolated from the other weather stations of Northern Europe. This station has a high correlation with Angmagssalik, in Southern Greenland. All three stations in the southwestern part of Greenland are also highly correlated, as well as the two meteorological stations in Eastern Canada. The weather stations of Southern Greenland and Eastern Canada, with the rare exception to Angmagssalik, have negative correlations with the group of 10 weather stations in Northern Europe, which is clearly seen in the bluish color. However, the values of these correlations are quite low for cases when they are statistically significant.

The patterns of spatial interconnection found in January turned out to be similar in February and March. The results for these months are shown in the Appendix A (Figures A1 and A2).

During the winter season, the SAT is especially dependent on atmospheric circulation regimes, and the NAO largely drives air temperature variations in Northern Europe, Greenland and North America. It is known that the positive phase of the NAO is associated with enhanced westerlies and northward displacement of the jet stream, which determines the trajectories of cyclones. Therefore, during this phase, warm air masses have a greater effect on Northern Europe, whereas Greenland and Eastern Canada experience negative temperature anomalies. During the negative phase, the orientation of the jet stream becomes close to West–East, and as a result, the cyclones pass by Northern Europe.

In addition, the westerlies are significantly weakened, which gives rise to more frequent Arctic air outbreaks. As a consequence, lower temperatures are observed over Northern Europe, whereas Greenland and Eastern Canada have higher temperatures [26–32].

Therefore, high correlation coefficients among the weather stations in Northern Europe are explained by the same weather processes generated by the large-scale climate patterns, and the NAO plays a dominant role. Similarly, negative correlations of the group of weather stations located in Southern Greenland and Eastern Canada with the group of weather stations in Northern Europe are explained by the peculiarities of the atmospheric circulation during a particular phase of the NAO.

### 3.2. Results of the SAT Time Series Modeling for the Present Period 1951–2016 in January

#### 3.2.1. Performance of the Non-Stationary Models

Based on the described methodology, calculations of the characteristics of stationary and non-stationary models were carried out for each weather station. The results are presented in Table 2, where the bold font highlights effective differences from the model of the stationary sample at $\Delta \geq 10\%$, the corresponding year of stepwise changes ($T_{step}$) and statistically significant values of the correlation coefficients for the linear trends (R). Based on the geographical location and correlation analysis, the weather station Reykjavik is excluded from the group of weather stations in Northern Europe.

**Table 2.** The results of the efficiency assessment of non-stationary models for the SAT time series at every weather station in January (1951–2016 period with the size of the time series n = 66). The bold font for $\Delta_{tr.}$ and $\Delta_{step}$ is used when the values are higher than or equal to 10%. The bold font for $T_{step}$ is used for the cases when $\Delta_{step} \geq 10\%$. The bold font for R denotes statistically significant correlation coefficients of linear trends for the whole period.

| Weather Stations | $\Delta_{tr.}$, % | $\Delta_{step}$, % | $F_{tr.}$ | $F_{step}$ | $T_{step}$ | R |
|---|---|---|---|---|---|---|
| SCHEFFERVILLE | 0.0 | 1.3 | 1.00 | 1.03 | 2006 | 0.01 |
| FORT CHIMO | 0.2 | 2.5 | 1.00 | 1.05 | 2006 | 0.06 |
| EGEDESMINDE | 1.5 | 7.5 | 1.03 | 1.17 | 1997 | 0.17 |
| GODTHAB NUUK | 0.0 | 2.0 | 1.00 | 1.04 | 1983 | −0.02 |
| PRINS CHRISTI | 0.9 | 4.8 | 1.02 | 1.10 | 2000 | 0.14 |
| ANGMAGSSALIK | 5.3 | 9.3 | 1.12 | 1.22 | 1998 | **0.32** |
| REYKJAVIK | 4.2 | 7.6 | 1.09 | 1.17 | 1996 | **0.29** |
| VARDO | 3.8 | 8.3 | 1.08 | 1.19 | 1991 | **0.27** |
| BODO VI | 4.4 | **11.7** | 1.09 | 1.28 | **1988** | **0.29** |
| ORLAND III | 4.2 | **11.9** | 1.09 | 1.29 | **1988** | **0.29** |
| BERGEN | 2.3 | 7.3 | 1.05 | 1.16 | 1988 | 0.21 |
| ABERDEEN | **11.8** | **16.6** | 1.29 | 1.44 | **1989** | **0.47** |
| KOBENHAVN | 2.8 | 7.9 | 1.06 | 1.18 | 1988 | **0.24** |
| OSTERSUND | 5.5 | **12.8** | 1.12 | 1.32 | **1988** | **0.33** |
| JYVASKYLA | 0.5 | 4.5 | 1.01 | 1.10 | 1988 | 0.10 |
| St. PETERBURG | 2.3 | 7.2 | 1.05 | 1.16 | 1988 | 0.21 |
| ARHANGEL'SK | 0.0 | 3.3 | 1.00 | 1.07 | 1963 | 0.03 |

Four non-stationary models in the form of stepwise changes were found to be effective for the weather stations in Northern Europe, three of which are the stations with mainly marine influence. Although Fisher statistics were slightly lower than required for statistical significance, their values were coherently increased for the cases with $\Delta \geq 10\%$. In addition, just looking at the numbers for every weather station, one should note a better performance of the model of stepwise changes compared to the linear trend model, because $\Delta_{step}$ is always higher than $\Delta_{tr}$. At almost all weather stations with effective non-stationary models, the year of stepwise changes was 1988, and, in one case, it was in 1989.

3.2.2. Graphical Representation of the Model of Stepwise Changes for the Weather Stations in Northern Europe, and Assessment of the Warming

To illustrate the model of stepwise changes, linear trends were calculated for both parts of the time series without overlapping (Figure 3).

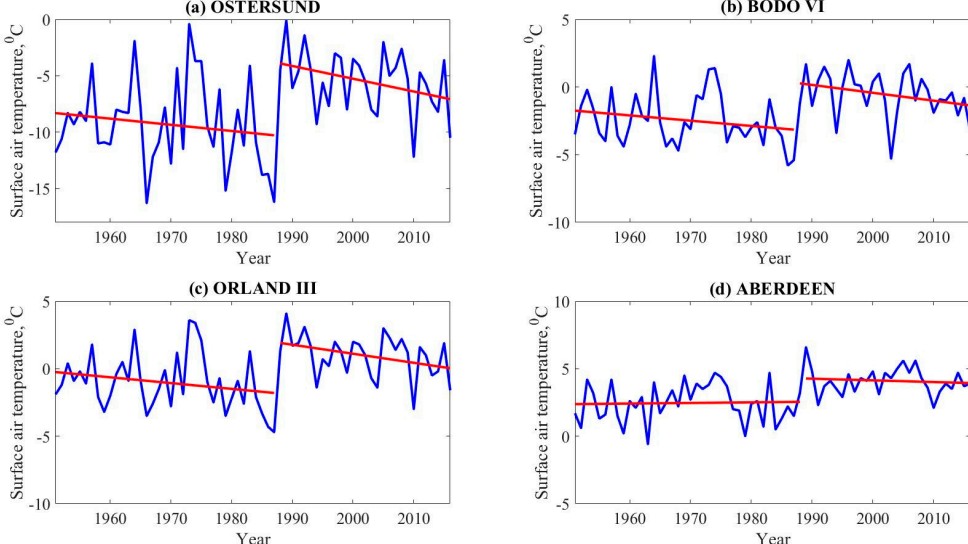

**Figure 3.** The model of stepwise changes for the SAT in January at the weather stations, (**a**) Ostersund, (**b**) Bodo VI, (**c**) Orland III and (**d**) Aberdeen.

In all cases with Δ ≥ 10%, the trends of both parts of the time series were found to be statistically not significant, which is consistent with the model of stepwise changes. Nevertheless, at all weather stations, except Aberdeen, the trends were negative, which is likely due to high variability in January.

The differences in the mean temperatures for the weather stations with the years of stepwise changes in 1988, 1989 and 1991 are summarized in Table 3. The weather station Jyvaskyla is not included, because $\Delta_{step}$ is only 4.5% for this station. The bold font highlights the weather stations with $\Delta_{step} \geq 10\%$.

**Table 3.** Assessment of the warming in Northern Europe based on the divided trends (1951–2016 period, January). The weather stations with the bold font are those with $\Delta_{step} \geq 10\%$.

| Year of Stepwise Changes | Weather Station | Difference in the Mean SAT, °C |
|:---:|:---:|:---:|
| 1991 | VARDO | 1.26 |
| 1988 | **BODO VI** | 1.90 |
| 1988 | **ORLAND III** | 2.00 |
| 1988 | BERGEN | 1.43 |
| 1989 | **ABERDEEN** | 1.65 |
| 1988 | KOBENHAVN | 1.72 |
| 1988 | **OSTERSUND** | 3.80 |
| 1988 | St. PETERBURG | 2.93 |

As shown, the differences in mean temperature are quite high for all weather stations, thus confirming the rapid warming. Moreover, the condition $\Delta_{step} \geq 10\%$ does not necessarily lead to the highest differences. This means that, from the physical point of view, the year of stepwise changes around 1988 is additionally justified. According to the two-sample Student's t-test, the differences are statistically significant. The mean difference for the weather stations in January is 2.09 °C.

*3.3. Results of the SAT Time Series Modeling for the Present Period 1951–2016 in February*

3.3.1. Performance of the Non-Stationary Models

The results of the SAT time series modeling in February are shown in Table 4. The bold font is used in the same way as for January.

**Table 4.** The results of the efficiency assessment of non-stationary models for the SAT time series at every weather station in February (1951–2016 period, with the size of the time series n = 66). The bold font for $\Delta_{step}$ is used when the values are higher than or equal to 10%. The bold font for $T_{step}$ is used for the cases when $\Delta_{step} \geq 10\%$. The bold font for R denotes statistically significant correlation coefficients of linear trends for the whole period.

| Weather Stations | $\Delta_{tr.}$, % | $\Delta_{step}$, % | $F_{tr.}$ | $F_{step}$ | $T_{step}$ | R |
|---|---|---|---|---|---|---|
| SCHEFFERVILLE | 1.4 | 3.6 | 1.03 | 1.08 | 1961 | −0.16 |
| FORT CHIMO | 0.4 | 2.2 | 1.01 | 1.05 | 1961 | −0.09 |
| EGEDESMINDE | 0.7 | 9.2 | 1.01 | 1.21 | 2003 | 0.11 |
| GODTHAB NUUK | 0.5 | 3.6 | 1.01 | 1.08 | 1971 | −0.10 |
| PRINS CHRISTI | 0.1 | 7.1 | 1.00 | 1.16 | 2004 | 0.03 |
| ANGMAGSSALIK | 3.6 | 7.5 | 1.08 | 1.17 | 2003 | **0.27** |
| REYKJAVIK | 0.3 | 4.4 | 1.01 | 1.09 | 2003 | 0.08 |
| VARDO | 4.8 | 7.5 | 1.10 | 1.17 | 1988 | **0.31** |
| BODO VI | 3.3 | 4.7 | 1.07 | 1.10 | 1972 | **0.25** |
| ORLAND III | 5.0 | 5.6 | 1.11 | 1.12 | 1971 | **0.31** |
| BERGEN | 3.5 | 5.6 | 1.07 | 1.12 | 1988 | **0.26** |
| ABERDEEN | 9.5 | **11.9** | 1.22 | 1.29 | **1988** | **0.43** |
| KOBENHAVN | 5.3 | **10.1** | 1.12 | 1.24 | **1988** | **0.32** |
| OSTERSUND | 4.9 | 6.2 | 1.11 | 1.14 | 1989 | **0.31** |
| JYVASKYLA | 2.5 | 4.8 | 1.05 | 1.10 | 1988 | 0.22 |
| St. PETERBURG | 4.9 | 7.6 | 1.11 | 1.17 | 1987 | **0.31** |
| ARHANGEL'SK | 1.1 | 2.5 | 1.02 | 1.05 | 1988 | 0.15 |

As in January, the dominant year of stepwise changes at the weather stations in Northern Europe was 1988. However, the time series for only two weather stations were in very good agreement with the non-stationary models; namely, the models of stepwise changes. Nevertheless, at least two more weather stations in Northern Europe nearly satisfied the condition of a 10% difference (Vardo and St. Peterburg). In addition, the year of stepwise changes during 1987–1989 was determined for eight of the 10 weather stations in Northern Europe identified in the correlation analysis, whereas, in January, the consistency was nine out of 10, if the Vardo station is included with the year 1991.

3.3.2. Graphical Representation of the Model of Stepwise Changes for the Weather Stations in Northern Europe, and Assessment of the Warming

In all cases, the trends of each part of the time series were not statistically significant. However, after the years of stepwise changes at the weather stations Kobenhavn and St. Peterburg, the trends were negative. This repeats the situation of January, albeit with weaker trends. Again, this is likely due to high wintertime variability.

An illustration of the model of stepwise changes is shown in Figure 4.

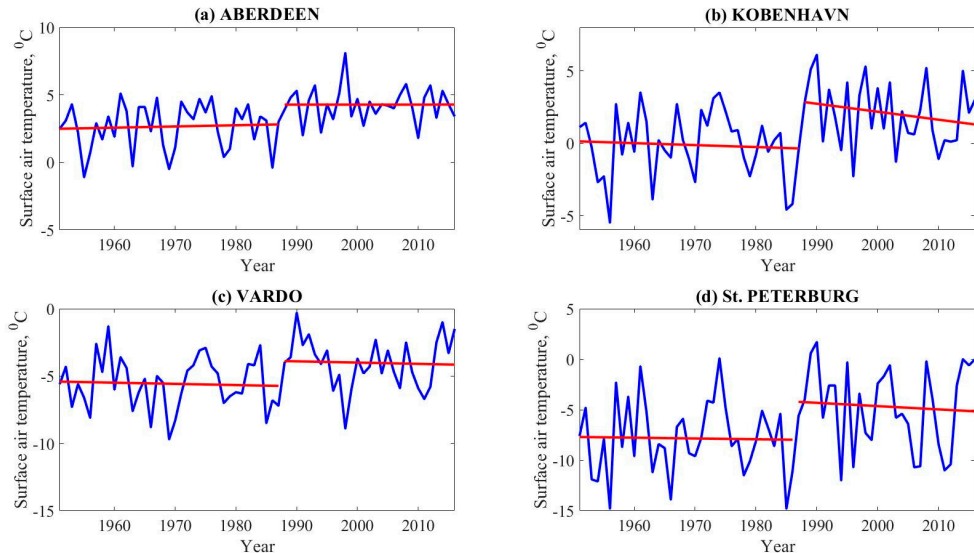

**Figure 4.** The model of stepwise changes for the SAT in February at the weather stations, (**a**) Aberdeen, (**b**) Kobenhavn, (**c**) Vardo and (**d**) St. Peterburg.

The weather stations Aberdeen and Kobenhavn represent the cases where $\Delta_{step} \geq 10\%$, and the stations Vardo and St. Peterburg are for cases where the differences are closest to 10% (7.5% and 7.6%, respectively).

The differences in the mean temperatures for the weather stations with the years of stepwise changes in 1987–1989, where the corresponding model is effective or close to the condition of 10%, are summarized in Table 5. The bold font highlights the weather stations with $\Delta_{step} \geq 10\%$.

**Table 5.** Assessment of the warming in Northern Europe based on the divided trends (1951–2016 period, February). The weather stations with the bold font are those with $\Delta_{step} \geq 10\%$.

| Year of Stepwise Changes | Weather Station | Difference in the Mean SAT, °C |
|:---:|:---:|:---:|
| 1988 | VARDO | 1.55 |
| 1988 | **ABERDEEN** | 1.63 |
| 1988 | **KOBENHAVN** | 2.18 |
| 1989 | OSTERSUND | 2.72 |
| 1987 | St. PETERBURG | 3.14 |

As shown, the differences in mean temperatures are quite high for all weather stations, thus confirming the rapid warming in February. Again, the condition $\Delta_{step} \geq 10\%$ does not necessarily lead to the highest differences. It should be emphasized that these temperature differences cannot be directly compared with those in January, because the stationarity in February is significantly higher than that in January. However, one can conclude that warming in February has a similar magnitude to that of January. According to the two-sample Student's t-test, the differences are statistically significant. The mean difference for the weather stations in February is 2.24 °C.

*3.4. Results of the SAT Time Series Modeling for the Present Period 1951–2016 in March*

3.4.1. Performance of the Non-Stationary Models

The results of the SAT time series modeling in March are shown in Table 6. The bold font is used in the same way as for January and February; however, the weather stations of Northern Europe for which the year of stepwise changes is 1989, where the corresponding model is close to being effective, are also highlighted.

**Table 6.** The results of the efficiency assessment of non-stationary models for the SAT time series at every weather station in March (1951–2016 period with the size of the time series n = 66). The bold font for $\Delta_{tr.}$ and $\Delta_{step}$ is used when the values are higher than or equal to 10%. The bold font for $T_{step}$ is used for the year 1989 when $\Delta_{step}$ is close to 10%. The bold font for R denotes statistically significant correlation coefficients of linear trends for the whole period.

| Weather Stations | $\Delta_{tr.}$, % | $\Delta_{step}$, % | $F_{tr.}$ | $F_{step}$ | $T_{step}$ | R |
|---|---|---|---|---|---|---|
| SCHEFFERVILLE | 1.1 | 2.5 | 1.02 | 1.05 | 1963 | −0.15 |
| FORT CHIMO | 0.1 | 2.9 | 1.00 | 1.06 | 1963 | −0.05 |
| EGEDESMINDE | 2.5 | **12.9** | 1.05 | 1.32 | 2003 | 0.22 |
| GODTHAB NUUK | 0.7 | 2.8 | 1.01 | 1.06 | 1964 | −0.12 |
| PRINS CHRISTI | 0.0 | 4.1 | 1.00 | 1.09 | 2001 | −0.02 |
| ANGMAGSSALIK | 3.3 | 9.1 | 1.07 | 1.21 | 2003 | **0.26** |
| REYKJAVIK | 0.1 | 5.1 | 1.00 | 1.11 | 2003 | 0.04 |
| **VARDO** | 7.4 | 7.7 | 1.17 | 1.17 | **1989** | **0.38** |
| BODO VI | 3.5 | 4.3 | 1.07 | 1.09 | 1989 | **0.26** |
| ORLAND III | 2.7 | 3.3 | 1.06 | 1.07 | 2003 | 0.23 |
| BERGEN | 2.5 | 5.1 | 1.05 | 1.11 | 1989 | 0.22 |
| **ABERDEEN** | 7.0 | 9.5 | 1.16 | 1.22 | **1989** | **0.37** |
| **KOBENHAVN** | 7.4 | 9.4 | 1.17 | 1.22 | **1989** | **0.38** |
| **OSTERSUND** | 4.5 | 6.5 | 1.10 | 1.14 | **1989** | **0.30** |
| JYVASKYLA | 5.3 | 6.4 | 1.12 | 1.14 | 1967 | **0.32** |
| St. PETERBURG | **10.0** | **13.7** | 1.24 | 1.34 | 1966 | **0.44** |
| ARHANGEL'SK | 4.1 | 8.8 | 1.09 | 1.20 | 1967 | **0.28** |

These results show that, in Northern Europe, the non-stationarity of the time series in March is the least pronounced among the three considered months. Furthermore, in March, the year of stepwise changes for the weather stations in Northern Europe is less stable: a number of weather stations had years of stepwise changes in the late 1960s and in 2003, with one station even being effective in terms of the non-stationarity. Overall, the consistency of the year of the stepwise jump in 1989 is six out of 10. Thus, from January to March, the consistency of the year of the stepwise jump during 1987–1989 for the group of 10 weather stations in Northern Europe decreases from 90% to 80%, and then to 60%.

One of the weather stations in Southern Greenland has a difference higher than 10%; however, this result is not consistent with the previous months, and cannot be attributed to a particular group of weather stations, which is the case for Northern Europe.

The four weather stations highlighted in bold are considered further.

### 3.4.2. Graphical Representation of the Model of Stepwise Changes for the Weather Stations in Northern Europe, and Assessment of the warming

In all cases, the trends of each part of the time series are not statistically significant, which is consistent with the model of stepwise changes. The situation in March is different from February and January, as there are no negative trends. Thus, from January to March, there is a gradual change in trends consistent with the increasing stationarity of the time series towards March.

An illustration of the model of stepwise changes is shown in Figure 5.

The trends are shown for the four weather stations of Northern Europe with the year of stepwise jump in 1989, and that have $\Delta_{step}$ very close to 10%. The results show the temperature jump upwards after 1989, but it is not so pronounced as in January and February. The trends of both parts of the time series are nearly zero, which represents a classical example of the model of stepwise changes.

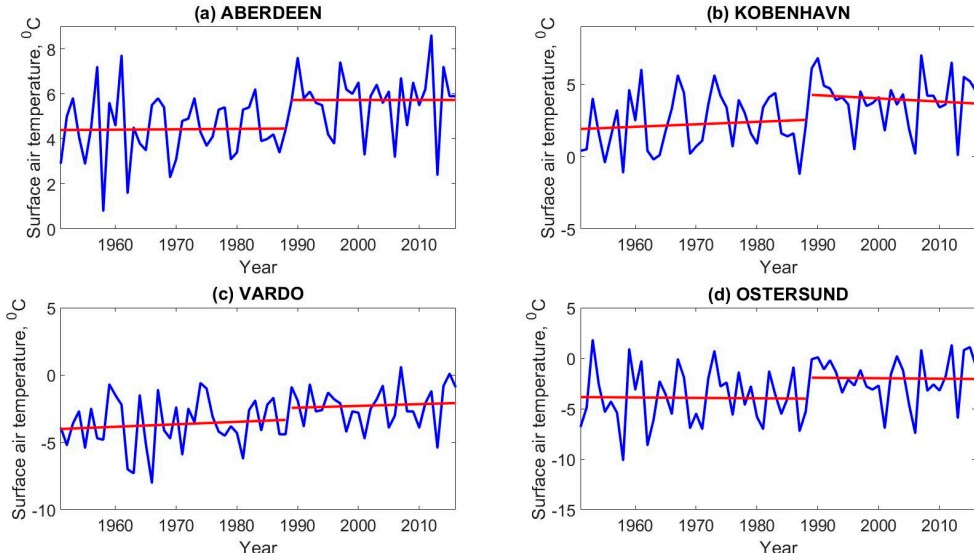

**Figure 5.** The model of stepwise changes for the SAT in March at the weather stations (**a**) Aberdeen, (**b**) Kobenhavn, (**c**) Vardo and (**d**) Ostersund.

The differences in the mean temperatures for the weather stations with the year of stepwise changes in 1989 are summarized in Table 7.

**Table 7.** Assessment of the warming in Northern Europe based on the divided trends (1951–2016 period, March).

| Year of Stepwise Changes | Weather Station | Difference in the Mean SAT, °C |
|:---:|:---:|:---:|
| 1989 | VARDO | 1.41 |
| 1989 | ABERDEEN | 1.30 |
| 1989 | KOBENHAVN | 1.74 |
| 1989 | OSTERSUND | 1.93 |

As shown, the differences in mean temperatures are generally lower in March compared to January and February, and the number of suitable stations is also lower. These are the consequences of the highest stationarity in March among the three months. However, the warming in March is also documented, although its magnitude is lower than that of January and February. According to the two-sample Student's t-test, the differences are statistically significant. The mean difference for the weather stations in March is 1.60 °C.

*3.5. Results of the SAT Time Series Modeling for the Previous Period 1872–1950*

For some weather stations in Northern Europe, long SAT time series exist, which allow us to compare the results for the present period with the earlier one. Thus, it can be determined whether the anomalies of the current period are a common feature of climate variability that could be observed previously. The chosen stations are the same as those used to determine the differences in the mean SAT; however, their number is lower because of the limited time series length.

Table 8 presents the results of the SAT time series modeling for the weather stations in Northern Europe in January, February and March.

**Table 8.** The results of the efficiency assessment of non-stationary models for the SAT time series at the weather stations in Northern Europe in January, February and March (1872–1950 period with the size of the time series n = 79). The bold font for R denotes statistically significant correlation coefficients of linear trends for the whole period.

| January | | | | | | |
|---|---|---|---|---|---|---|
| **Weather stations** | $\Delta_{tr.},\%$ | $\Delta_{step},\%$ | $F_{tr.}$ | $F_{step}$ | $T_{step}$ | **R** |
| ABERDEEN | 0.1 | 2.4 | 1.00 | 1.05 | 1939 | 0.04 |
| BERGEN | 0.0 | 2.2 | 1.00 | 1.04 | 1940 | 0.03 |
| BODO VI | 0.1 | 3.2 | 1.00 | 1.07 | 1940 | −0.03 |
| KOBENHAVN | 0.1 | 2.9 | 1.00 | 1.06 | 1940 | 0.05 |
| St. PETERBURG | 0.0 | 2.3 | 1.00 | 1.05 | 1940 | 0.00 |
| VARDO | 2.1 | 4.5 | 1.04 | 1.10 | 1922 | 0.20 |
| February | | | | | | |
| **Weather stations** | $\Delta_{tr.},\%$ | $\Delta_{step},\%$ | $F_{tr.}$ | $F_{step}$ | $T_{step}$ | **R** |
| ABERDEEN | 0.5 | 2.2 | 1.01 | 1.04 | 1903 | 0.10 |
| KOBENHAVN | 0.2 | 1.1 | 1.01 | 1.02 | 1903 | 0.07 |
| St. PETERBURG | 0.0 | 0.4 | 1.00 | 1.01 | 1902 | 0.01 |
| VARDO | 0.2 | 1.4 | 1.00 | 1.03 | 1920 | 0.07 |
| March | | | | | | |
| **Weather stations** | $\Delta_{tr.},\%$ | $\Delta_{step},\%$ | $F_{tr.}$ | $F_{step}$ | $T_{step}$ | **R** |
| ABERDEEN | 3.5 | 4.1 | 1.07 | 1.09 | 1920 | **0.26** |
| KOBENHAVN | 1.5 | 2.8 | 1.03 | 1.06 | 1893 | 0.17 |
| VARDO | 1.2 | 3.6 | 1.03 | 1.08 | 1903 | 0.16 |

As shown, the SAT time series for the period from 1872 to 1950 reflect the stationary model. Unlike the present period, there is no any statistical significance, thus confirming the unprecedented climate change from 1951 to 2016. The present climate change is pronounced, and has a significant non-stationary component.

The existence of stationarity during 1872–1950, when there was both a cold period before the 1920s [2] and a subsequent warm period (early twentieth-century warming, or ETCW), might indicate that these anomalous periods offset each other. For the period 1951–2016, there was also a cold period during the 1960–1970s [2,57] and a subsequent warm period; however, in this case, the non-stationarity might indicate that the warming dominates. This is consistent with the fact that the present-day warming has a more global pattern compared to the relatively local ETCW. Nevertheless, the northernmost weather station, Vardo, has a signal of that warming in January and February, because the years of stepwise changes were determined as 1920 and 1922, and $\Delta_{step}$ was the highest for January compared to all other cases.

### 3.6. Results of the SAT Time Series Modeling for the Entire Period 1872–2016

For additional confirmation of the robustness of the SAT jump during 1987–1989 in Northern Europe, the modeling was also carried out for the whole period 1872–2016.

Table 9 presents the results of the SAT time series modeling for the weather stations in Northern Europe in January, February and March.

**Table 9.** The results of the efficiency assessment of non-stationary models for the SAT time series at the weather stations in Northern Europe in January, February and March (1872–2016 period with the size of the time series n = 145). The bold font for R denotes statistically significant correlation coefficients of linear trends for the whole period.

| | | | January | | | |
|---|---|---|---|---|---|---|
| **Weather stations** | $\Delta_{tr.},\%$ | $\Delta_{step},\%$ | $F_{tr.}$ | $F_{step}$ | $T_{step}$ | **R** |
| ABERDEEN | 0.0 | 3.0 | 1.00 | 1.06 | 1989 | −0.02 |
| BERGEN | 1.0 | 3.5 | 1.02 | 1.07 | 1988 | 0.14 |
| BODO VI | 0.0 | 2.8 | 1.00 | 1.06 | 1989 | 0.01 |
| KOBENHAVN | 3.2 | 5.7 | 1.07 | 1.12 | 1988 | **0.25** |
| St. PETERBURG | 1.7 | 4.7 | 1.03 | 1.10 | 1988 | **0.18** |
| VARDO | 2.7 | 3.9 | 1.06 | 1.08 | 1991 | **0.23** |
| | | | February | | | |
| **Weather stations** | $\Delta_{tr.},\%$ | $\Delta_{step},\%$ | $F_{tr.}$ | $F_{step}$ | $T_{step}$ | **R** |
| ABERDEEN | 0.0 | 2.4 | 1.00 | 1.05 | 1988 | 0.02 |
| KOBENHAVN | 3.2 | 6.4 | 1.07 | 1.14 | 1988 | **0.25** |
| St. PETERBURG | 2.7 | 5.6 | 1.06 | 1.12 | 1987 | **0.23** |
| VARDO | 2.0 | 4.6 | 1.04 | 1.10 | 1988 | **0.20** |
| | | | March | | | |
| **Weather stations** | $\Delta_{tr.},\%$ | $\Delta_{step},\%$ | $F_{tr.}$ | $F_{step}$ | $T_{step}$ | **R** |
| ABERDEEN | 3.9 | 5.4 | 1.08 | 1.12 | 1989 | **0.28** |
| KOBENHAVN | 7.0 | 7.7 | 1.16 | 1.17 | 1989 | **0.37** |
| VARDO | 9.4 | 9.1 | 1.22 | 1.21 | 1972 | **0.42** |

For all months, the years with the stepwise jumps were also during 1987–1989, with only one mismatch in March. In January, an exception was the northernmost weather station, Vardo, with the year 1991, which was also the case for the shorter, present period 1951–2016. The only small difference was in the one-year shift for the station Bodo VI, which was 1989 here and 1988 for the period 1951–2016. In February, the years of stepwise changes perfectly corresponded to the situation for the present period. In March, the years with the stepwise jumps were also during 1987–1989, except for the weather station Vardo, where this year was 1972, compared with the year 1989 found in the present period.

Although the consistency in March was lower than that in January and February, the robustness of the identified jumps was also confirmed. Similar to January and February, with the exception of the weather station Vardo, the differences from the model of the stationary sample are lower than for the period 1951–2016, which also indicates that the rapid changes started to occur during the present period, whereas the earlier period 1872–1950 had stable conditions.

*3.7. Assessment of the Coherence of Changes between the NAO and SAT at the Weather Stations in Northern Europe during January, February and March*

The temperature field is largely formed depending on a particular atmospheric circulation regime. Therefore, in this section, the role of the key climate index characterizing atmospheric circulation is considered. As was previously mentioned in the 'Materials and Methods' section, the NAO index was considered. In order to be consistent with the most frequent years of stepwise jumps in the SAT, the trends were calculated for the periods 1951–1987 and 1988–2016 in January and February, and 1951–1988 and 1989–2016 in March.

In Figure 6, the NAO trends for all months are shown.

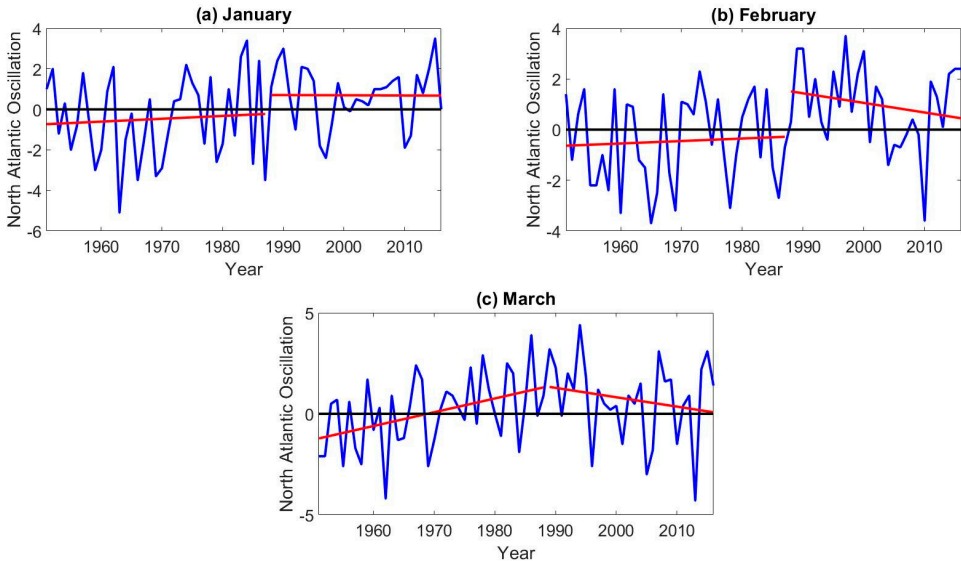

**Figure 6.** The North Atlantic Oscillation (NAO) trends in (**a**) January, (**b**) February and (**c**) March.

During all months, the frequency of the NAO positive phase was significantly higher after the late 1980s, which is clearly visible relative to the reference zero line, and confirmed by the trends located in the domain of the positive phase. The only statistically significant trend was for the first part of the time series in March.

Thus, a jump of the SAT in the late 1980s can be explained by a change in the pattern of the NAO index. Subsequently, the dominant occurrence of the positive phase of the oscillation led to the regime with higher SATs in Northern Europe. There was a pronounced coherence of changes in the NAO index and SAT.

The differences in the mean NAO during the considered months are summarized in Table 10. The numbers in bold are statistically significant differences according to the two-sample Student's t-test.

**Table 10.** Assessment of the mean NAO changes based on the divided trends (1951–2016 period, all months). The bold values in the second column indicate statistically significant differences in the mean NAO according to the two-sample Student's t-test.

| Month | Difference in the Mean NAO |
|---------|:---:|
| January | **1.17** |
| February | **1.44** |
| March | 0.66 |

As shown, the differences were statistically significant in January and February, and, from the physical point of view, these changes were also high for this climate index. A lower change without statistical significance in March is consistent with the highest stationarity among the three considered months for the SAT time series and the lowest SAT differences for this month.

The results derived from an application of the same modeling approach as for the SAT time series are shown in Table 11.

**Table 11.** The results of the efficiency assessment of non-stationary models for the NAO time series in January, February and March (1951–2016 period with the size of the time series n = 66). The bold font for R denotes statistically significant correlation coefficients of linear trends for the whole period.

| Month | $\Delta_{tr.}$, % | $\Delta_{step}$, % | $F_{tr.}$ | $F_{step}$ | $T_{step}$ | R |
|---|---|---|---|---|---|---|
| January | 4.2 | 6.7 | 1.09 | 1.15 | 1972 | **0.29** |
| February | 5.3 | 8.1 | 1.12 | 1.18 | 1988 | **0.32** |
| March | 3.0 | 6.7 | 1.06 | 1.15 | 1966 | **0.24** |

The NAO drives the SAT; therefore, one should expect the identification of the jumps in the NAO during the same years as in the SAT, or earlier. Generally, the results correspond to that reasoning; however, the differences in the stepwise model from the model of the stationary mean were lower than 10%. In this regard, the best coherence was achieved in February because, in this month, the year of stepwise changes was 1988, and the respective model was very close to being effective. In January, the significantly earlier year of stepwise changes could be identified because of the high wintertime variability, which was probably also the reason for the negative SAT trends in this month. In March, the year 1966 is consistent with the fact that, in this month, a number of weather stations in Northern Europe had years of stepwise changes in 1966 and 1967. In addition, it is important to note that, during all three months, the correlation coefficient of the linear trend was positive and statistically significant.

However, these results should be considered with caution, for several reasons. First, although the construction of the stepwise model is technically applicable to any time series, it has physical limitations. The SAT is a predicand, and the NAO is a predictor. The applied method is diagnostic in nature; therefore, the detection of the jump is physically more correct for the dependent variable (SAT) than for the variable that drives the changes (NAO). In addition, there are other models that are not covered in this study, e.g., various spectral models, which can be advantageous when considering large-scale modes of variability.

Nevertheless, the division of the NAO time series based on the years of stepwise changes in the SAT time series provides an objective understanding that the NAO has significantly contributed to the SAT increase in Northern Europe. This was confirmed by the trends located in the positive NAO phase, significant differences in means and even partly by the stepwise model applied to the NAO.

## 4. Discussion and Conclusions

This study emphasized the severe patterns of the present climate change in the North Atlantic region. Model projections indicate that the Earth will experience increasing heat extremes with continued warming [11,58,59]. At the same time, despite the global warming effect, some regions may have temperatures lower than normal, as is already the case for some parts of Eurasia (the warm Arctic–cold Eurasia pattern) [60,61]. This means that, in addition to warming, the instability of the climate system might lead to more frequent polar air outbreaks and warm air intrusions with a net warming effect, thus introducing discontinuities. These events are considered to be a result of the sea ice reduction in the Arctic because of the warming; there is evidence for a wavier jet stream pattern in response to the warming Arctic [62–64]. The results in this paper are in good agreement with the mentioned consequences of a changing climate. Indeed, the rapid changes were confirmed, together with the increased instability of the climate system when there is a pronounced discontinuity of warming coherent with a shift in the NAO index. A similar shift has already been documented [31,32,65], and the findings in this study confirm its real effects and physical significance.

In summary, the following conclusions can be drawn from this research:

- Climatic changes were assessed in January, February and March for the present period (1951–2016) and the longer previous period (1872–1950). A comparison of the results confirmed the unprecedented nature of the current climate change in Northern Europe, because the non-stationarity of the time series was much more pronounced during 1951–2016 compared to

1872–1950. However, the time series for the period 1951–2016 were stationary in Eastern Canada during all months. Moreover, there was no evidence for changes similar to those in Northern Europe in Southern Greenland, although the winter climate in Southern Greenland is not as stable as that of Eastern Canada.

- Statistical modeling of time series was carried out, which, for cases with non-stationary time series, revealed the suitability of the model of stepwise changes at the weather stations in Northern Europe.

- At most weather stations in Northern Europe, a stepwise jump of the SAT was observed in 1987–1989. However, in March, such stability was conserved only for the weather stations located near the ocean. The robustness of the jump was confirmed by the identification of the same years of stepwise jumps for the entire period 1872–2016. This was also supported by correlation analysis, which showed that monthly SATs of a large group of 10 weather stations in Northern Europe were spatially interconnected.

- The most pronounced and unstable climatic changes in Northern Europe were observed in January as, in this month, the highest number of non-stationary models were effective.

- From January to March, there was an increase in the stationarity of the time series at the weather stations in Northern Europe, and the temperature difference between the two parts of the time series was the lowest in March. Overall, the average wintertime warming due to the stepwise SAT jump was about 2 °C.

- The NAO was in good agreement with changes in the SAT, which were confirmed by the divided trends. The interrelation of this climate index with the SAT, especially during the winter season, is very well known; however, in this study, it was additionally demonstrated that the NAO has significantly contributed to the SAT jump in Northern Europe as one of the driving mechanisms.

This study can be elaborated upon by the investigation of other months, seasonal averages and annual means. In addition, some improvements regarding the statistical limitations presented in this paper are advisable in future research. It would be of interest to utilize other techniques and analyze the time series based on gridded products that can be obtained from re-analyses and climate models. This would allow the assessment of whether the persistence and robustness of the climatic changes found in the observations are present in the datasets, which incorporate modeling techniques and different scenarios of forcings.

Another important issue worth addressing in future research is the determination of the precise physical mechanisms and related dynamic processes that led to a transition of the winter climate state to a new one in Northern Europe during 1987–1989. Although the NAO has significantly contributed to the SAT jump, other factors are also clearly involved, and the NAO is interrelated with other climate indices. For the North Atlantic region, it would be especially relevant to investigate the AMO and AMOC indices in the framework of the new findings obtained in this research. Moreover, novel modeling studies are needed to provide an exhaustive explanation of the causes of the climate shift.

**Author Contributions:** Conceptualization, M.M.L.; methodology, M.M.L. and V.A.L.; software, M.M.L. and V.A.L.; validation, M.M.L., V.A.L. and I.L.B.; formal analysis, M.M.L.; investigation, M.M.L.; resources, M.M.L. and V.A.L.; data curation, M.M.L. and V.A.L.; writing—original draft preparation, M.M.L.; writing—review and editing, V.A.L. and I.L.B.; visualization, M.M.L.; supervision, V.A.L. and I.L.B.; project administration, I.L.B.; funding acquisition, M.M.L. and I.L.B. All authors have read and agreed to the published version of the manuscript.

**Funding:** This research was funded by RFBR, project number 19-35-90083.

**Acknowledgments:** M.M.L. is grateful to the support of the Nansen Scientific Society (Bergen, Norway). The authors are thankful to reviewers and editors whose evaluation helped to improve the quality of the manuscript.

**Conflicts of Interest:** The authors declare no conflict of interest. The funder had no role in the design of the study; in the collection, analyses, or interpretation of data; in the writing of the manuscript, or in the decision to publish the results.

## Appendix A

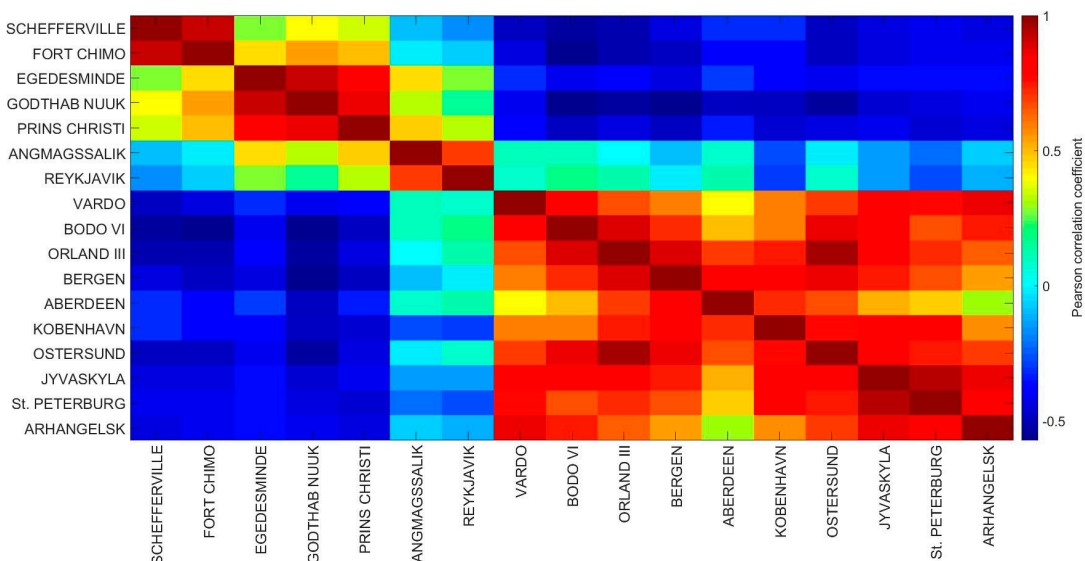

**Figure A1.** Pearson correlation coefficients among all weather stations in February (1951–2016 period).

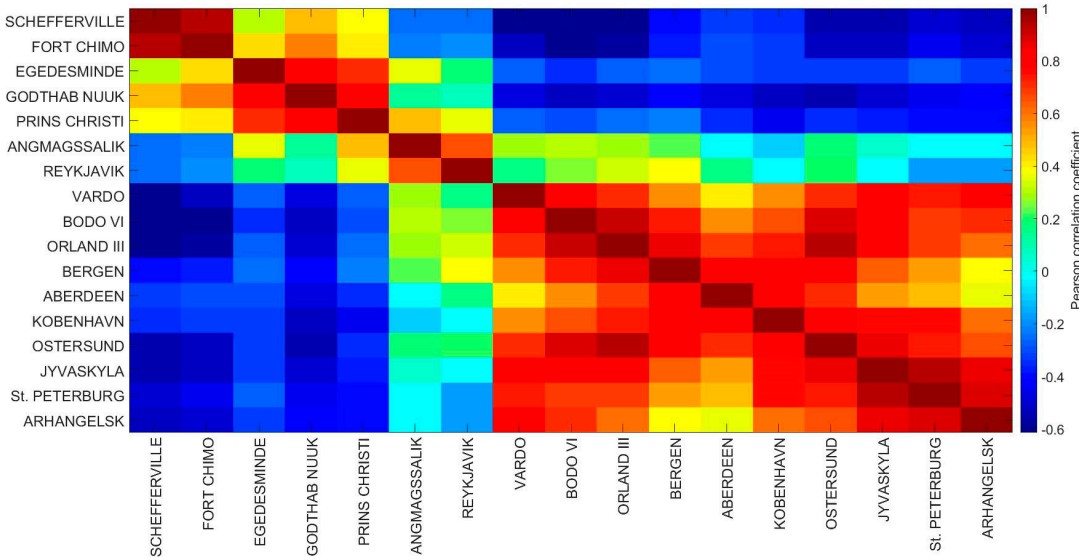

**Figure A2.** Pearson correlation coefficients among all weather stations in March (1951–2016 period).

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
