# Peer review of "Discontinuities in Wintertime Warming in Northern Europe during 1951–2016"

_climate, doi:10.3390/cli8060080_

Round 1

Reviewer 1 Report

Comments in the attached file.

Reviewer 2 Report

The authors presented an interesting study that uses the SAT from several weather stations located at the northern part of the North Atlantic Ocean to represent their main trends and their links with NAO.

The manuscript is interesting and the results are relevant. However, among the mathematical analysis performed, which is very detailed, I was expecting to find some explanations about the physical links and the dynamical mechanisms that can explain the obtained results. I suggest to the authors to try to include this explanation in this manuscript.

The manuscript could be also improved if the authors include a deep explanation of the climatological conditions of the area under study. For example, I miss the link with ocean phenomena, and particularly with the Gulf Current, the North Atlantic surface branch of the thermohaline circulation (THC). There is a really strong link between NAO and THC and THC and surface temperatures in this region.

I also recommended to include a stronger justification of why this work is needed. Saying that “Despite numerous studies on their role in the regional and global climate, more reliable information about their particular effects on climate tendencies is needed.” is not enough.

Some of the selected meteorological stations present very long datasets. The authors known if that data was uniformed considering eventually changes in the equipment, in the location of the meteorological stations, or in the data processing among the years?

Why NCEP/NCAR? There are a lot of reanalysis data with better resolution, as, for example, ERA5.

I am not an English native speaker, but I recommend a revision of the text. Long sentences are presented, and sometimes the manuscript is difficult to follow.

Other minor comments:

The reference to the softwares used to produce the figures is not needed. The authors can eliminate the sentence: “(created using the GIS software MapInfo). All other Figures in this manuscript are created using a language of technical computing and a high-level programming language MATLAB.”.

In the sentence: “As seen, the SAT and climate indices are essential components and drivers of the present climate change” I suggest to change “components and drivers” by “indicators that can be used to monitor and follow the present climate change”

Round 2

Reviewer 1 Report

Even if in my previous review I recommended rejection, I acknowlege the significant improvement of the current version of the manuscript, especially due to the  MDPI’s English editing service. Now the manuscript is more clear, more readable and the scientific soundness has been highlighted. I found that the authors have also addressed the other comments, so I recommend publication. 

Reviewer 2 Report

After the revision, the manuscript can be accepted in the present form